# Absolute Quantification of Selected microRNAs Expression in Endometrial Cancer by Digital PCR

**DOI:** 10.3390/ijms25063286

**Published:** 2024-03-14

**Authors:** Anna Bogaczyk, Natalia Potocka, Sylwia Paszek, Marzena Skrzypa, Alina Zuchowska, Michał Kośny, Marta Kluz, Izabela Zawlik, Tomasz Kluz

**Affiliations:** 1Department of Gynecology, Gynecology Oncology and Obstetrics, Fryderyk Chopin University Hospital, 35-055 Rzeszow, Poland; annabogaczyk@interia.pl (A.B.); jtkluz@interia.pl (T.K.); 2Laboratory of Molecular Biology, Centre for Innovative Research in Medical and Natural Sciences, Medical College of Rzeszow University, 35-959 Rzeszow, Poland; npotocka@ur.edu.pl (N.P.); spaszek@ur.edu.pl (S.P.); mskrzypa@ur.edu.pl (M.S.); 3Institute of Medical Sciences, Medical College of Rzeszow University, 35-959 Rzeszow, Poland; aowsiak@ur.edu.pl; 4Department of Hematology, Medical University of Lodz, 90-419 Łódź, Poland; mkosny1@gmail.com; 5Department of Pathology, Fryderyk Chopin University Hospital, 35-055 Rzeszow, Poland; marta.kluz@interia.pl

**Keywords:** miRNA, endometrial cancer, miR-21-5p, miR-205-5p, miR-222-3p, reference gene, SNORD48, U6

## Abstract

MicroRNAs (miRNA) are involved in the process of carcinogenesis, including the development of endometrial cancer (EC). This study aimed to investigate the association between the expression of three miRNAs (miR-21-5p, miR-205-5p, and miR-222-3p) in endometrial cancer tissues. In addition, the stability of expression of SNORD48 and U6, which were initially planned to be used as reference miRNAs for normalization, was investigated. Endometrial tissue was obtained from 111 patients with EC during hysterectomy and from 19 patients undergoing surgery for uterine fibroids or pelvic organ prolapse as a control group without neoplastic changes. Our study was based on calculations made with a digital PCR method (Qiagen, Hilden, Germany) to measure the absolute expression. In the endometrial cancer tissue, miR-205-5p was upregulated, while miR-222-3p and SNORD48 were downregulated compared to the control group. We detected statistically significant correlation of miR-205-5p, U6, and SNORD48 expression with different histological grades; the expression of miR-205-5p increases with the histopathological grade advancement (intraepithelial neoplasia- EIN = 1590, G1 = 3367.2, G2 = 8067 and G3 = 20,360), while U6 and SNORD expression decreases from EIN to G2 and increases again in the G3 grade (U6: EIN = 19,032, G1 = 16,482.4, G2 = 13,642.4, G3 = 133,008; SNORD48: EIN = 97,088, G1 = 59,520, G2 = 43,544, G3 = 227,200). Our study suggests that upregulation of miR-205-5p and downregulation of miR-222-3p and SNORD48 may influence development of endometrial cancer. Moreover, miR-205-5p, U6, and SNORD48 expression changes may be associated with progression of endometrial cancer. The results also indicate that SNORD48 and U6, commonly used as internal references, may influence endometrial cancer development and progression; therefore, they should not be used as references. However, it is important to note that further research is required to understand their role in endometrial cancer.

## 1. Introduction

### 1.1. Endometrial Cancer

Endometrial cancer is one of the most common malignant tumors in women in developed countries and ranks fifth in morbidity after breast cancer, colorectal cancer, lung cancer, and thyroid cancer [1]. The cancer occurred in 319,600 cases of global incidence in 2012 and is on the rise [2]. In 2018, there were 382,069 new cases of endometrial cancer and 89,929 deaths [3]. In 2020, there were 417,367 new cases and 97,370 deaths due to endometrial cancer worldwide [1]. The presented figures show a clear upward trend in both morbidity and mortality due to endometrial cancer.

From the clinicopathological point of view, the endometrial cancer is divided into two types: type I and type II (according to Bokhman’s dualistic theory).

Type I is the endometrioid type (EEC) characterized by a state of hyperestrogenism and occurs mainly in obese women with polycystic ovary syndrome (PCOS), and anovulatory cycles. It can also occur in women with a genetic predisposition (e.g., Lynch syndrome). Type I EC has a favorable prognosis and accounts for 80–90% of endometrial cancers. Type II ECs include serous, clear cell, and undifferentiated carcinomas, and are associated with higher patient age, and high stage and grade, and have a poor prognosis [4].

In 2013, a new molecular classification of endometrial cancer was introduced by The Cancer Genome Atlas (TCGA), which indicates a paradigm shift from morphological to molecular classification. The TCGA identified four molecular subgroups characterized by:-*POLE* mutation (POLEmut group) (Polymerase Epsilon) (7%);-Microsatellite instability (MSI group) which results from mismatch repair deficiency (MMRd) (28%);-High somatic copy number changes (driven by the *TP53* mutation, also called the p53abn group) (26%);-Low copy number without defined molecular profile, no specific molecular profile (NSMP group) (39%) [5].

Each of these groups has a different prognosis. Tumors with the *POLE* mutation have an extremely favorable prognosis, while the group with a high copy number due to the *TP53* mutation has a poor prognosis. The prognosis for tumors with mismatch repair deficiency (MMRd) and no specific molecular profile (NSMP) is relatively favorable. Common mutations in the MSI group include *ARID5B*, *PTEN*, and mutations of the phosphatidylinositol-3 kinase family genes, including *PIK3CA* and *PIK3R1* [6].

The most common genetic mutation in endometrial cancer is the mutation of the *PTEN* suppressor gene (phosphatase and tensin homolog deleted from chromosome-10). Functional inactivation of *PTEN* is associated with the initiation and progression of endometrial cancer. Studies have shown a 34–55% frequency of somatic mutations of the *PTEN* gene with a 50–83% frequency of loss or a decline of the PTEN protein [7,8,9]. Khatami et al. showed that the reduction in *PTEN* gene expression in endometrial cancer tissues is caused by hypermethylation of its promoter. *PTEN* promoter methylation was observed by researchers in 52.0% of cancer tissues [10]. Another important gene in EC carcinogenesis is *SOX17*, which acts as a tumor suppressor gene, and its inactivation is important in endometrial cancer progression through EMT (epithelial–mesenchymal transition) regulation [11]. According to the recommendations of the Federation of Gynecology and Obstetrics (FIGO), endometrial cancer can also be divided into three grades depending on the degree of histological differentiation [G1–G3]. G1 is characterized by having ≤5% solid growth pattern, G2 has from 6 to 50% solid growth pattern, while G3 has >50% solid growth pattern.

As we wrote at the beginning of the introduction, the increasing number of new cases of EC and the increase in the number of deaths due to this cancer forces us to look for the causes of this phenomenon and new markers for the advancement of the cancer or the tendency to metastasize. MicroRNAs may act as prognostic factors. Therefore, in our study we focus on changes in the expression of three microRNAs (miR-21-5p, miR-205-5p, and miR-222-3p) in endometrial cells in patients with various degrees of histopathological differentiation. We hope that determining changes in the expression of our microRNAs will be the basis for improving the prognosis of patients with EC and extending the life of patients.

### 1.2. MicroRNA in Control of Gene Expression

According to the NIH Roadmap Epigenomics Project, the term epigenetics refers to both inherited changes in gene activity and expression (in the progeny of cells or individuals) and stable, long-term changes in the transcriptional potential of the cell. Epigenetic modifications are defined as arbitrary modifications in genomic DNA that result in transcriptional silencing and can be both beneficial and disadvantageous. For instance, in cancer, epigenetic modifications can cause high levels of activation of genes, so-called oncogenes, but can also have adverse effects by silencing tumor suppressor genes. Three different types of mechanisms are known to cause gene silencing: DNA methylation, histone modifications, and RNA-related silencing [12].

MiRNAs are particularly interesting molecules that contribute to the regulation of epigenetic gene expression. They were discovered in 1993 and are non-coding, single-stranded, small RNA molecules about 21–25 nucleotides long [13]. MiRNAs are important post-transcriptional regulators, and may act as oncogenes or tumor suppressors in the development of endometrial cancer [14,15]. In this study, we used digital PCR to analyze absolute expression of the following microRNAs: miR-21-5p, miR-205-5p, and miR-222-3p. Initially, SNORD48 and U6 were used as references for normalization. However, our results showed that they were unstable in endometrial cancer tissue. The miRNAs that were analyzed in the study are described below.

MiR-21 acts by directly targeting the 3′-UTR of *PTEN* (the phosphatase and tensin homolog) mRNA, which is a tumor suppressor gene. Functional inactivation of *PTEN* is associated with the initiation and progression of endometrial cancer [16].

Qin et al. found that miR-21 expression could be correlated with advanced clinical stages, deep myometrial invasion, and high histological grade [16].

Wang et al. showed that miR-21-5p promoted EMT (epithelial to mesenchymal transition), while miR-21-5p silencing had the opposite effect in endometrial cancer cells. EMT promotion by miR-21-5p may be mediated by targeting *SOX17* (SRY-box 17), and is correlated with poor survival in endometrial cancer patients. *SOX17* encodes a 414 amino acid protein member of the HMG-box (SOX) superfamily of transcription factors related to SRY. It should be mentioned that the epithelial to mesenchymal transition is an important process by which epithelial cells lose their character, e.g., they lose epithelial markers such as E-cadherin, and acquire mesenchymal characteristics, e.g., by an increase in mesenchymal markers such as fibronectin, which leads to tumor metastasis [11].

Tian et al. also studied miR-21-5p, and also confirmed its inhibitory effect on *PTEN*. In addition, researchers showed that miR-21-5p was a potential target of NBTA1 (Long noncoding RNA neuroblastoma-associated transcript 1), and was confirmed by a reporter assay of luciferase activity. The study confirmed through cellular experiments that NBTA1 was targeted to bind to miR-21-5p, and overexpression of miR-21-5p not only promoted EC proliferation and invasion but also inhibited apoptosis. It also had the effect of blocking NBTA1 on EC cells [17]. MiR-21-5p has also been studied in other tumors, e.g., peritoneal cancer [18], lung cancer [19], colorectal cancer [20], breast cancer [21], and ovarian cancer [22].

Studying the role and mechanisms of miR-21-5p in EC is important from both a scientific and medical point of view. Stopping cancer metastasis plays an important role in treating patients and extending their lives. It seems that miR-21-5p gives us such possibilities.

The mechanism of action of miR-205-5p regulates EMT by silencing ZEB2 and ZEB1 (homeobox 1, 2 binding the E-box to the zinc finger) [23]. MiR-205-5p has been shown to be overexpressed in head and neck cancer cell lines [24], cervical cancer [25], kidney and bladder cancer [26], non-small cell carcinoma lung cancer [27], and breast cancer [23], compared to normal tissues. Karaayvaz et al. indicated that miR-205-5p expression levels were significantly associated with endometrial cancer patient survival. Patients with low miR-205-5p expression tended to have better survival than patients with high miR-205-5p levels. This supported the observations that elevated levels of miR-205-5p in tumor tissues could lead to poorer survival rates in endometrial cancer patients [28].

Zhuo and Yu observed that miR-205-5p is more highly expressed in Ishikawa-PR cells compared to Ishikawa cells. In addition, the researchers observed that miR-205-5p inhibitor treatment significantly inhibited the growth of Ishikawa cells and PR tumor cells. These observations show that miR-205-5p may be involved in progesterone resistance in the EC, although the mechanism of progesterone resistance in the EC remains unclear. Researchers also showed a new perspective on the regulatory mechanism of miR-205-5p on the *PTEN* gene in PR-ECs, where *PTEN* levels were significantly reduced in Ishikawa’s PR cells, and the miR-205-5p inhibitor increased *PTEN* expression in these cells. These findings show us that miR-205-5p plays an important role in the loss of *PTEN* expression and PR development in EC cells [29]. It should be mentioned that *PTEN* is the most frequently modified gene in the EC, and is located on human chromosome 10 and encodes a tyrosine kinase. It is estimated that 34–83% of ECs and precancerous lesions show reduced expression of *PTEN* [30]. *PTEN* is a tumor suppressor, promotes apoptosis, and its deletion or mutation results in tumor development [29].

The expression of miR-205-5p was studied in gastric cancer and found to be decreased [31].

Lu et al. studied the effect of miR-205-5p on Paclitaxel resistance; they found that miR-205-5p was upregulated in EC tissues and targeted *FOXO1* (*FOXO1* is a transcription factor widely distributed in the heart, brain, lung, and other tissues and organs). MiR-205-5p induced an increase in Paclitaxel resistance, and contributed to EC cell neoplasia by affecting *FOXO1*. Knocking down miR-205-5p increased the sensitivity of EC cells to Paclitaxel, resulting in reduced cell proliferation and increased apoptosis. Researchers have shown a new possible direction for treating EC patients by regulating miR-205-5p [32].

It should also be mentioned that lncRNAs also have an effect on endometrial cancer by modulating the miR-205-5p-*PTEN* axis. Such an example was presented by Xin et al. describing the effect of lncRNA, i.e., LA16c-313D11.11, on the miR-205-5p-*PTEN* axis, where LA16c-313D11.11 can inhibit the development and progression of EC by acting like a miR-205-5p sponge, and thus thereby indirectly increasing *PTEN* expression [33]. Xin et al. reported that as many as 13 lncRNAs can be associated with the miR-205-5p-*PTEN* network and they are LINC00657, RP11-395G23.3, HNRNPU-AS1, MCM3AP-AS1, SNHG5, SNHG16, LA16c-313D11.11, THAP9-AS1, RP11-379K17.11, RP11-38P22.2, RP11-349A22.5, UBXN8, and ERVK3-1. Researchers concluded that these 13 lncRNAs could act as endogenous spongy RNAs to interact with and suppress miR-205-5p [34].

The role of miR-205-5p in EC still requires additional research and a better understanding of its impact on the mechanism of EC carcinogenesis.

MiR-222-3p has been studied in many types of cancer, including endometrial cancer. Liu et al. showed that miR-222-3p was upregulated in ERα-negative EC tissues. It has been also shown that overexpression of miR-222-3p was correlated with higher grades, later stages, and more nodal metastasis. Moreover, it has been found that, in vivo, miR-222-3p suppression could significantly inhibit tumor growth [35]. However, the role of the clinical significance of miR-222-3p in EC is still not fully understood and therefore it was selected for our analyses.

Fu et al. studied miR-222-3p expression in epithelial ovarian cancer (EOC) patients in mouse models and cell lines. They found it to be a suppressor, and higher expression of miR-222-3p was associated with better overall survival in EOC patients. In addition, they showed that its level was negatively correlated with tumor growth in vivo. Experiments conducted by the researchers in vitro showed that miR-222-3p inhibited the proliferation and migration of EOC cells and reduced AKT phosphorylation. *GNAI2* (G protein alpha inhibiting activity polypeptide 2, Galphai2, Giα2) has been identified as a target of miR-222-3p, and it promotes EOC cell proliferation and is an activator of the PI3K/AKT pathway [36].

The situation is different in non-small cell lung cancer. Chen and Li showed that the expression of miR-222-3p was significantly upregulated in these cancer tissues and cell lines, and that the miR-222-3p inhibitor reduced the activity of non-small cell lung cancer, i.e., reduced cell proliferation and increased cell apoptosis. Overexpression of miR-222-3p in non-small cell lung cancer cells promoted cell proliferation and reduced their apoptosis. Researchers demonstrated Bcl-2-binding component 3, which was the target gene of miR-222-3p, and its knockdown attenuated the regulatory effect of the miR-222-3p inhibitor on cell proliferation and apoptosis in non-small cell lung cancer cells [37].

SNORD48 (Small Nucleolar RNA, C/D Box 48) (RNU48) is one of many small nuclear RNAs designated as members of the SNORDs family whose expression is stable (at a constant level) in many tissues. Therefore, it is often used as a reference miRNA in research on miRNA expression level (apart from, e.g., U6, SNORD 44). References genes were studied by Torres et al., who analyzed the stability of reference genes in patients with endometrial cancer. They showed that RNU48 was one of the most stable genes in EC. Researchers using both NormFinder and geNorm (https://norm.btm.umed.pl) indicated that SNORD48 was optimal for normalizing qPCR data in EC tissues. SNORD48 was also expressed equally between normal and tumor samples [38]. In contrast, Bignotti et al. studied the stability of SNORD48 in ovarian cancer and showed that SNORD48 is stable between malignant and normal tissues in ovarian cancer patients. The investigators supported the use of SNORD48 as the best reference for relative quantification in expression studies in ovarian cancer [39]. However, Egidi et al. studied gene stability in urine sediment from prostate cancer patients and showed that the SNORD48 value exceeded the limit of acceptability [40]. Additionally, Lawror et al. studied endogenous SNORD48 in patients irradiated for prostate cancer. The researchers showed that expression levels were associated with a low coefficient of variation after irradiation (6 Gy). Scientists have noticed the effect of radiation on changes in endogenous SNORD48 [41].

Jurcevic et al., using NormFinder, showed that using the five most stable genes including U6 provided the best normalization [42]. In contrast, Lou et al. showed that U6 is unstable in various tumor tissues. The study focused on breast cancer tissues, in which U6 expression was higher compared to healthy tissue, and U6 expression was similarly higher in cancer cells than in mesenchymal cells. High U6 expression levels were higher in liver and intrahepatic bile duct cancer tissues compared with adjacent normal tissues. Researchers have shown that U6 expression and distribution show high variability among several human cell types [43].

The absolute concentration of the studied miRNAs was determined using digital PCR (dPCR). This method offers increased sensitivity and a better limit of detection (LOD) due to the small reaction volume, which increases the effective concentration of the target miRNA. Therefore, the LOD is less restrictive for dPCR than for quantitative PCR (qPCR). Furthermore, digital PCR allows for the absolute quantification of the target nucleic acid in the sample, as opposed to real-time PCR which only provides relative quantification. This leads to more precise and accurate quantification, especially for low-abundance targets.

### 1.3. The Aim of Study

The aim of this study was to investigate the association between the expression of three miRNAs (miR-21-5p, miR-205-5p, and miR-222-3p) in endometrial cancer tissues. We examined miRNA changes in relation to histopathological differentiation and comorbidities in patients with endometrial cancer. Furthermore, we investigated the stability of SNORD48 and U6 expression in endometrial cancer tissues.

## 2. Results

The characteristics of the study group, including the clinical features and lifestyle of patients with endometrial cancer, are presented in Table 1. Quantitative variables were compared and analyzed using the U Mann–Whitney test due to the relatively small control group and the presence of deviating data distributions from normal. No statistical significance was obtained for the following factors: age at menarche (first period), vaginal births, caesarean sections, miscarriages, and hyperthyroidism. Statistically significant factors are menopausal status (last menstrual period), hypertension, diabetes (DM), hypothyroidism, and BMI. A statistically significant difference was also obtained in the age of the study and control groups. This results from the inability to select patients in the appropriate age group, because the material was obtained from endometrial tissue collected during hysterectomy. Therefore, patients undergoing surgery for uterine prolapse or with fibroids were included in the study as a control group.

In the above patients, the absolute expression of the following miRNAs was tested: miR-21-5p, miR-205-5p, miR-222-3p, U6, and SNORD48 in endometrial tissues. Statistically significant results were obtained for miR-205-5p (*p* < 0.001), miR-222-3p (*p* = 0.013), and SNORD48 (*p* < 0.001) expression. Table 2 shows the absolute expression of the tested miRNAs in copies per µL.

In the miRNA expression data used for further analysis, a logarithmic transformation was performed to obtain a normal distribution of the data quantified by digital PCR. The mean logarithmic miRNA expression values were used to determine the fold change (FC). Furthermore, both univariate and multivariate random regression analyses were conducted.

The expression level miRNA in the tested endometrial cancer tissue and in the control group as logarithmic data are presented in Figure 1 and Table 3. The analyses revealed that out of the three miRNAs tested, miR-205-5p (FC 1.4, *p* < 0.001) was upregulated, while miR-222-3p (FC 0.94, *p* = 0.013) was downregulated. In addition, SNORD48 (FC 0.92, *p* < 0.001) showed statistically significant changes in expression. After adjusting the *p*-value using the Benjamin–Hochberg method, differences in miRNA concentration between the study group and control group for miR-205-5p and SNORD48 remained *p* <0.001; however, for miR-222-3p, adjusted *p* = 0.022 (U6 and mir-21-5p remained statistically insignificant).

In view of the fact that our study group differed significantly from the control group and both groups differed in age and comorbidities, the remaining analyses were performed by dividing the study group according to the degree of histopathological differentiation.

To investigate potential factors associated with the occurrence of EC, we performed a univariate logistic regression analysis of clinical variables and miRNA expression levels. It was found that age (OR 1.12, 95% CI: 1.06–1.20, *p* < 0.001), BMI, and hypertension were significant clinical factors associated with the development of EC. Among the miRNAs tested, two showed significant associations with endometrial cancer: miR-205-5p (OR 7.27, 95% CI: 2.96–22.4, *p* < 0.001) and miR-222-3p (OR 0.12, 95% CI: 0.02–0.53, *p* = 0.011). Additionally, the study yielded statistically significant results for SNORD48 (OR 0.04, 95% CI: 0.008 to 0.24, *p* < 0.001) (Table 4).

In the multivariate logistic regression analysis, only miR-205-5p (OR 6.05, 95%CI: 0.38–19.59, *p* = 0.003) was found to be associated with endometrial cancer. This association also persisted for SNORD48 (OR 0.01, 95%CI: 0.0003–0.47, *p* = 0.018) (Table 5).

In addition, an analysis was performed in which the degree of histological differentiation (grading) according to FIGO of endometrial cancer was considered and divided into three groups:-G1—highly differentiated cancer (<5% of solid tissue)—30 patients;-G2—moderately differentiated cancer (6–50% of the solid part)—47 patients;-G3—poorly differentiated cancer (>50% of the solid tissue)—12 patients.

The division also included EIN (intraepithelial neoplasia)—22 patients.

In the next stage, the expression level of miRNAs was determined in individual degrees of differentiation (G1, G2 and G3) and in EIN (Table 6).

The Kruskal–Wallis test was conducted to assess differences in the expression level of miRNAs and was determined in individual degrees of differentiation (G1, G2 and G3), EIN, and control group. The analysis revealed a significant overall difference in concentration of U6, SNORD48, and miR-205-5 between groups. Subsequent post hoc analyses were performed to investigate pairwise group differences. Results indicated that those pairs differed significantly: for U6, three out of ten pairs: G1 vs. G3 (*p* = 0.001), G2 vs. G3 (*p* = 0.002), and G3 vs. EIN (*p* = 0.022)); for SNORD48, two pairs out of ten: G1 vs. control group (*p* < 0.001) and G2 vs. control group (*p* < 0.001)); and for miR-205-5, five pairs out of ten: G1 vs. control group (*p* = 0.001), G2 vs. EIN (*p* = 0.006), G2 vs. control group (*p* < 0.001), G3 vs. EIN (*p* = 0.013), and G3 vs. control group (*p* < 0.001). For 222-3p, no statistical difference was seen in post hoc tests.

A univariate logistic regression analysis was conducted to examine the effect of the studied miRNAs on individual degrees of differentiation in endometrial cancer. No significant results were obtained for G1 and G2. The results of univariate logistic regression analysis for EIN and G3 are presented in Table 7 and Table 8. Statistical significance was achieved for miR-205-5p (OR 0.48) in EIN (Table 7) and miR-21-5p (OR 0.49), miR-205-5p (OR 3.05), U6 (OR 9.67), and SNORD48 (OR 0.48) for G3 (Table 8).

Since age was found to be statistically significant (*p* < 0.001), the expression of individual miRNAs in the endometrial cancer study group was determined depending on age (< and >50 years of age). We obtained a statistically significant result in the case of miR-205-5p (*p* = 0.041) (Table 9).

The study revealed a 1.23-fold increase in the risk of endometrial cancer with increasing body mass index (BMI), as determined by univariate logistic regression analysis (*p* = 0.001) (Table 4). Therefore, in patients with endometrial cancer, we analyzed the relationship between individual miRNA expression and BMI. Patients were grouped by BMI (Table 10):-18.5–24.9—normal weight (14 patients);-25–29.9—overweight (43 patients);-30 and above means obesity (54 patients).

In the study group, only 13% had normal body weight, and as many as 49% of patients were obese.

The study group was divided depending on the differentiation of the tumor (Table 11). In the EIN group, 86% of patients had abnormal body weight; in the G1 group, 90%; in the G2 group, 89.3%; and in the G3 group, 81.8%.

Then, the absolute expression of individual miRNAs was determined depending on BMI in the study group. However, we did not obtain statistically significant results (Table 12).

Then, the expression of individual miRNAs was determined in the endometrial cancer study group depending on the presence of comorbidities such as hypertension (Table 13), diabetes (Table 14), and hypothyroidism (Table 15). No statistically significant results were obtained.

The study also assessed miRNA stability analysis (reference miRs U6 and SNORD48) using NormiRazor. NormiRazor is a tool that implements three different existing normalization algorithms—geNorm, NormFinder, and BestKeeper. The originally estimated U6 and SNORD48 were found to be unstable. Stability analysis was performed on single miRNAs and a combination of two miRNAs (U6 and SNORD48). However, none of the miRNAs tested, used singly or in combination, are stable enough in endometrial cancer tissue to be applied as a reference miRNA (Table 16). These results show that both U6 and SNORD48 are not good reference miRNAs for endometrial cancer.

## 3. Discussion

Our study included 130 patients, with 111 in the study group and 19 in the control group. All patients had endometrial material collected during hysterectomy. Our control group consisted of endometrial tissues collected from patients undergoing surgery due to pelvic organ prolapse or uterine fibroids.

This study observed changes in miRNA expression levels in endometrial cancer tissue. In particular, miR-205-5p was upregulated, while miR-222-3p was downregulated, in comparison to the control group. In contrast, altered miR-21-5p expression was only significant in histological group G3.

In the study we observed increased expression of miR-205-5p in the endometrial cancer study group compared to the control group (FC 1.4, *p* < 0.001). Furthermore, we confirmed the association of elevated miR-205-5p expression in patients with endometrial cancer through univariate (OR 7.72, *p* = 0.037) and multivariate (OR 6.05, *p* = 0.003) logistic regression analyses. The levels of miRNAs were analyzed in different histopathological differentiation groups (EIN, G1, G2, and G3). It was observed that miR-205-5p is overexpressed in patients diagnosed with EIN (*p* = 0.018) and in stage G3 endometrial cancer (*p* = 0.037). This observation is very important because we know that miR-205-5p causes a decrease in the expression of *PTEN*, which is a tumor suppressor gene and thus may influence carcinogenesis [29]. MiR-205-5p can also increase cancer cell proliferation and reduce apoptosis by affecting *FOXO1*. Understanding these mechanisms in more detail may have an impact on inhibiting carcinogenesis in EC through the use of miR-205-5p inhibitors [32].

Additionally, Xin W. et al. demonstrated a significant increase in miR-205-5p expression in endometrial cancer tissues. This study was based on endometrial tissue material obtained from 60 patients (30 patients constituted the study group and 30 patients constituted the control group). The control group included, similarly to our study, women treated for diseases not related to endometrial tissue, i.e., uterine fibroids and pelvic organ prolapse. In our study, the tissue material was obtained by hysterectomy or curettage of the uterine cavity only during hysterectomy [34]. Kulinczak et al. demonstrated higher expression of miR-205-5p not only in EC tissue, but also in tissue adjacent to the tumor. This study was based on a 49-person study group and a 25-person control group [44].

Most studies suggest that miR-222-3p has oncogenic effects on cancer cells [35,37,45]. The selection and inclusion of miR-222-3p in the study was due to its potential impact on EC development [35]. Previous studies show that miR-222-3p has an oncogenic effect by regulating MMP2 (matrix metalloproteinaza), MMP9, *TRPS1* (trichorhinophalangeal syndrome type 1), and CDKN1C/p57 (cyclin-dependent kinase inhibitor 1C) [35,46,47]. Our observations do not confirm the oncogenic role of miR-222-3p and put this microRNA in a new light. Our study revealed that miR-222-3p was downregulated in endometrial cancer tissues (FC 4.74, *p* = 0.013). In contrast, significant changes in miR-222-3p expression levels were not demonstrated among the different histopathological classification groups. However, it is interesting to note that the levels of miR-222-3p tend to decrease across all groups, with the lowest expression observed in G3. Our study suggests a rather suppressive role of miR-222-3p and offers new therapeutic possibilities. By inducing an increase in the expression of miR-222-3p, we may be able to inhibit the proliferation of EC cells.

Another study on endometrial cancer showed an increase in miR-222-3p expression, but in ERα-negative tissues, while miR-222-3p expression was significantly lower in ERα-positive tissues. In this study, the expression level of miR-222-3p was inversely correlated with the expression of ERα, and the expression level of miR-222-3p was lower in lower grade tumors. The study was conducted on a group of 75 patients [35].

However, a completely different role of miR-222-3p was demonstrated by Fu et al. in a study conducted on a group of 74 patients with ovarian cancer. In this study, miR-222-3p acts as a suppressor of ovarian cancer. Researchers found longer survival in ovarian cancer patients with high levels of miR-222-3p, compared to a group of patients with low levels of miR-222-3p. This study shows that miR-222-3p may be a better prognostic indicator for ovarian cancer patients [36].

MiR-21-5p is a well-described PTEN inhibitor [16]; researchers have also described its promoting effect on EMT [11], which is why we decided to choose this oncogenic microRNA for our studies. However, no significant differences in miR-21-5p expression were observed in the EC study group compared to the control group.

Additionally, there were no statistically significant changes in miR-21-5p expression levels for any degree of histopathological differentiation compared to the control group. However, when analyzing the individual histopathological groups, a statistically significant increase in miR-21-5p expression was observed in histopathological differentiation G3 compared to the other classification groups (OR = 0.49, *p* = 0.019).

Although this miRNA is one of the most consistently expressed miRNAs in almost all types of human cancer, and may be a useful clinical biomarker and therapeutic target [18,19,22], it also increases in endometrial cancer. Sato et al. described high miR-21-5p expression in EC cells, and it was associated with greater disease progression and lymph node metastases [48].

Bouziyane et al. demonstrated the high diagnostic efficacy of miR-21-5p in their research. This study included a large group of 71 endometrial cancer tissues, 53 adjacent tissues, and 54 benign lesions [49].

In our study, we did not obtain a significant miR-21-5p result in EIN using univariate logistic regression, but a statistically significant result appeared in G3 (*p* = 0.019). Similar observations were reported by Sato et al. Sato et al. examined the relationship between miR-21-5p expression and clinicopathological characteristics. The study was conducted on a large number of patients with uterine corpus cancer; the entire group consisted of 230 patients, and there were 176 endometroid carcinomas (G1 = 77, G2 = 51, G3 = 48). Researchers showed high expression of miR-21-5p in cancer cells with a higher histological grade (G3 vs. G1, G2) in the case of endometrial cancer, *p* < 0.0001. However, they did not show any relationship between miR-21-5p expression and patient age (<60 vs. ≥60) or FIGO stage (stage I/II vs. III/IV) in their study [39].

Comorbidities are well-known risk factors for endometrial cancer and have been described previously [48,50,51,52,53]. The above data were analyzed using univariate logistic regression. As a result of this analysis, we confirmed that the patients’ age, BMI, and hypertension influence the incidence of endometrial cancer. The study group with endometrial cancer was divided according to the presence of comorbidities or without comorbid diseases. A statistically significant result was obtained in the group divided by patient age (*p* = 0.041). In this group, the expression of miR-205-5p was upregulated in the group >50 years of age. We did not obtain statistically significant results in the groups with hypertension, diabetes, or hypothyroidism.

In the knowledge that obesity is a significant risk factor for endometrial cancer [54], we also included this parameter in our study. As a result of our analysis, we confirmed that BMI is a risk factor for endometrial cancer (*p* < 0.001). It was concluded that there is a positive correlation between an increase in BMI and the risk of endometrial cancer (OR 1.23, *p* = 0.001). Furthermore, we analyzed the impact of BMI on the expression of the studied miRNAs in patients with endometrial cancer. However, we did not find any significant differences in miRNA expression between EC patients with a normal BMI and those who were overweight or obese. In EC patients with a higher-than-normal BMI, there was a trend for decreased miR-21-5p expression, although this result did not reach statistical significance. However, obesity and changes in miRNAs were described in previous works as a cancer risk factor [55]. Rodrigues et al. also showed increased miR-21-5p expression in obese individuals with hepatocarcinogenesis [56].

Our research shows that SNORD48 is not stable and its expression differs significantly in tissues from patients and in tissues from healthy people. This study showed that SNORD48 expression was downregulated in endometrial cancer tissue compared to the control group (FC 0.92, *p* < 0.001). Furthermore, SNORD48 was identified as a factor in the development of endometrial cancer in both univariate (OR 0.04, *p* < 0.01) and multivariate (OR 0.01, *p* = 0.018) logistic regression analyses. Furthermore, an analysis of the study group divided into histopathological grades revealed changes in SNORD48 expression. There was a higher expression of SNORD48 in the G3 group compared to the other histopathology groups, including EIN. The relationship was statistically significant (*p* < 0.001). Furthermore, we demonstrated the instability using the NormiRazor tool.

To date, the data on the effects of SNORD48 on endometrial cancer are limited. However, some studies suggest a potential link between SNORD48 and cancer. Egidi et al. studied urinary miRNAs in prostate cancer patients and SNORD48 exceeded the limits of acceptability [40].

Shen et al. examined SNORDs in colorectal cancer patients and observed upregulation of SNORDs 48 [57]. Rapti et al. used SNORD48 as a reference gene in colorectal cancer [58]. Bignotti et al. used SNORD48 in a study of serous undifferentiated ovarian cancer and it emerged as the best reference gene [39]. Mase et al. assessed the stability of five reference genes (U6, SNORD48, SNORD44, miR-16, and 5S) in atrial tissues of 18 cardiac surgery patients in sinus rhythm and atrial fibrillation. All quantitative stability methods ranked SNORD48 as the best reference gene [59].

In conclusion, our studies suggest that SNORD48 may influence the development of endometrial cancer. However, further research is needed to better understand the role of SNORD48 in the development of endometrial cancers.

The second miRNA that we planned to use as a reference is U6 (RNU6-1). It is recommended to use this miRNA as a control for endometrial cancer. Jurcevic et al. examining endogenous control genes in a rat model of endometrial cancer, and detected significant differences in U6 expression between malignant and non-malignant samples (*p* < 0.05) [42].

In this study, no statistically significant changes in U6 expression were observed between the study and control groups. However, the differences in U6 expression in different degrees of histological differentiation were unexpected, indicating its instability. The expression of this miRNA decreased in the early phases (EIN, G1, and G2), whereas it increased in G3 compared to the control group (*p* < 0.001). Furthermore, we assessed the stability of U6 in endometrial cancer tissue using the NormiRazor tool. The analysis confirmed the low stability of this miRNA. Therefore, we could not use U6 as a reference gene. The explanation of U6 expression changes in different histological grades requires further research.

## 4. Materials and Methods

### 4.1. Tissues Samples

The study involved 111 patients diagnosed with endometrial cancer. All diagnoses were confirmed by previous histopathological tests. These patients came to the Department of Gynecology, Gynecology Oncology and Obstetrics of the Fryderyk Chopin University Hospital in Rzeszów between 03/2021 and 09/2022 to start oncological treatment. The study also included 19 healthy women (without endometrial cancer) operated on in the local clinic due to pelvic organ prolapse or uterine fibroids, and they formed a control group. All women consented to the use of tissues for genetic testing Consent of the Bioethics Committee of the District Medical Chamber of 21 May 2020, resolution No. 54/B/2020. None of the patients received hormonal therapy, radiotherapy, or chemotherapy before sample collection.

### 4.2. MiRNA Isolation from Tissue Samples

The endometrial cancer tissue samples were frozen at −80 °C and stored in RNAprotect Tissue Reagent (Qiagen, Hilden, Germany). Isolation of total RNA, including miRNA, from cancer tissue was performed using the miRNeasy Mini Kit (Qiagen, Hilden, Germany) according to the manufacturer’s protocol. Initially, tissues thawed on ice were transferred to tubes containing 700 µL of QIAzol Lysis Reagent and then homogenized by sonication. After the addition of chloroform and centrifugation (15 min at 12,000× *g* at 4 °C), the upper aqueous phase containing total RNA, including miRNA, was mixed in new tubes with ethanol and then transferred to an RNeasy MiniElute Spin Column and washed. The RWT washing buffer was diluted in isopropanol. During the purification procedure, DNA digestion was performed using the RNase-free DNase kit (Qiagen, Hilden, Germany). A quantity of 30 µL of RNase-free water was used to elute the RNA. The RNA concentration was measured using the NanoDrop™ 2000c Spectrophotometer (ThermoFisher Scientific, Waltham, MA, USA), diluted to the final RNA concentration of 5 ng/μL, and then used. The quality of the isolated RNA was also assessed by electrophoretic separation on a 1% agarose gel. The RNA was reverse transcribed immediately after isolation.

### 4.3. Reverse Transcriptase Reaction and dPCR Method

MiRNAs were polyadenylated using poly(A) polymerase and reverse transcribed into cDNA using oligo-dT primers, which have a degenerate 3’ anchor that allows miRNA amplification in a real-time PCR reaction. Polyadenylation and reverse transcription were conducted in parallel in the same tube. CDNA was synthesized according to the manufacturer’s guidelines using the miRCURY LNA Reverse Transcription Kit (Qiagen, Hilden, Germany). The reactions were performed in a final volume of 10 µL, using 10 ng of RNA. The RT reaction was incubated at 42 °C for 60 min and inactivated at 95 °C using a T100™ 96-well thermocycler (Bio-Rad, Hercules, CA, USA). The cDNA was stored at −20 °C until further use.

The study used dPCR to quantify absolute expression of selected miRNAs expression by combining the limiting dilution, PCR endpoint, and Poisson statistics. QIAcuity System dPCR (Qiagen, Hilden, Germany) was used to determine the absolute expression of the miRNAs tested. The digital PCR (dPCR) technique uses microfluidic nanoplate technology. This method allows the quantification of nucleic acids based on measuring the fluorescence endpoint of each partition. The absolute expression for the miRNAs was determined: hsa-miR-21-5p, hsa-miR-205-5p, hsa-miR-222-3p, U6 snRNA (hsa, mmu), SNORD48 (hsa) using miRCURY LNA miRNA PCR Assay applying a Nanoplate 8.5 K 96-well (Qiagen, Hilden, Germany). An optimization was performed before the assays to select an appropriate concentration of cDNA and avoid partition overload. The optimization was conducted separately for each miRNA by analyzing a range of dilutions from 10-fold to 100-fold relative to the starting material. For dPCR reactions, miR-205-5p and miR-222-3p were used at a 20× dilution, while miR-21-5p, SNORD48, and U6 were used at a 40× dilution. A 12 µL reaction mixture was prepared, consisting of 4 µL Eva Green Master Mix (3×), 1.2 µL miRCURY LNA PCR primers (10×), and 3 µL of diluted cDNA. The samples were loaded into the wells of a 96-well PCR nanoplate, which was then sealed with an OIAcuity nanoplate seal. Thermal cycling conditions were as follows: 95 °C for 2 min, then 45 cycles of 95 °C for 15 s and 60 °C for 1 min, and three final steps at 4 °C for 5 min. The plate was read in the green channel using the parameters of an exposure time of 500 ms and a gain of 6. In our study, the lowest detectable values were for miR-205-5p, 32.5 copies/µL; miR-222-3p, 819.2 copies/µL; miR-21-5p, 56.96 copies/µL; SNORD48, 665.6 copies/µL; and U6, 1034.72 copies/µL. Initially, snRNAs U6 and SNORD48 were selected as potential reference genes. Due to the lack of stability in the SNORD48 and U6 reference gene samples, we present the results only as absolute expression (copies/µL). The calculations were performed using QIAcuity software version 2.1.8 (Qiagen, Hilden, Germany).

Figure 2 shows the workflow for the experiment and analysis of the data.

### 4.4. Statistical Analysis

Statistical analyses were performed for the following subgroups: tissue, tissue (study group only), tissue (control group only).

MiRNA expression quantified by dPCR was logarithmically transformed to obtain a normal distribution of the data. MiRNA stability analysis was then assessed using NormiRazor. NormiRazor is a tool that implements three different existing normalization algorithms—geNorm, NormFinder, and BestKeeper. The originally estimated U6 and SNORD48 snRNAs were found to be unstable; therefore, the data were not normalized by these miRNAs. This approach results in higher log values for higher miRNA expression, allowing simple interpretation of biomarkers. Patient data and differential expression analysis for quantitative variables was performed using the independent *t* test, Welch’s *t* test, and Mann–Whitney U or Kruskal–Wallis test depending on the distribution of variables and equality of variances tested by the Shapiro–Wilk test and Levene’s test, respectively. For qualitative variables, the chi2 test was used with Yate’s correction and Fisher’s exact test when needed, depending on sample size. Univariate logistic regression analysis was used to assess the impact of individual factors on the occurrence of endometrial cancer. Subsequently, statistically significant variables identified in the univariate analysis were included in a multivariate logistic regression model. Nominal variables are presented as numbers and percentages or as median (range or interquartile range) and mean (SD), depending on the normality of their distribution. Statistical analysis was performed using Statistica 13.1 (Tibco, Palo Alto, CA, USA) and R version 4.2.1. *p*-values less than 0.05 were considered significant.

## 5. Conclusions

Our study was based on calculations performed by digital PCR to determine the expression of the studied miRNAs in patients with endometrial cancer. Expression changes in endometrial cancer tissue were observed for miR-205-5p, miR-222-3p, and SNORD48. In the endometrial cancer tissue, miR-205-5p was upregulated compared to the control group, while miR-222-3p and SNORD48 were downregulated. We also observed an association between miR-205-5p, U6, and SNORD48 expression levels and histological grades. The expression level of U6 varied in different stages of EC, suggesting its lack of stability. We found no association between the expression of the miRNAs studied and body mass index or other comorbidities in patients with endometrial cancer.

In this study, SNORD48 and U6 did not meet the expectations of a reference miRNA because they may also influence the development of endometrial cancer, suggesting that they should not be used as reference miRNAs in endometrial cancer. However, further research is needed to better understand the role of SNORD48 and U6 in the development of endometrial cancers.

Our research requires confirmation in a larger group of patients and an increase in the number of miRNAs tested. It creates new diagnostic possibilities and, through new targeted therapies, it gives us new treatment possibilities.

### Limitations

Our study had some limitations. The first was the control group, which consisted of only 19 patients.

There was also an uneven distribution of patients in particular groups of histopathological differentiation—the EIN group consisted of 22 patients, the G1 group consisted of 30 patients, the G2 group consisted of 47 patients, while the G3 group consisted of only 12 patients.

## Figures and Tables

**Figure 1 ijms-25-03286-f001:**
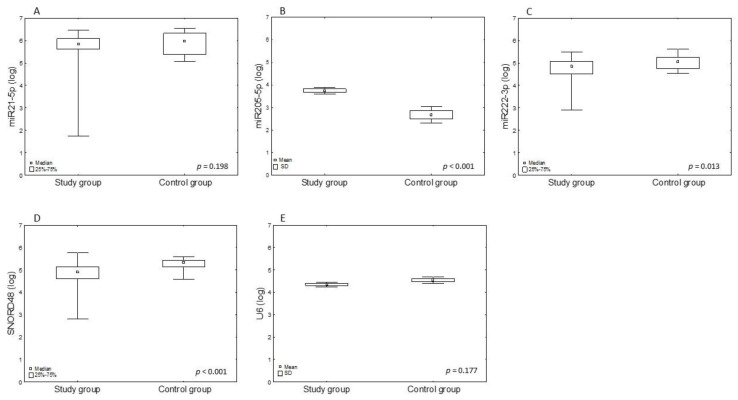
The graphs show logarithmically transformed miR-21-5p (**A**), miR-205-5p (**B**), miR-222-3p (**C**), U6 (**D**), and SNORD48 (**E**) expression in endometrial cancer tissues quantified by dPCR, where it was shown that the expression of miR-205-5p in endometrial cancer tissue increased compared to the control group, and the expression of miR-222-3p decreased.

**Figure 2 ijms-25-03286-f002:**
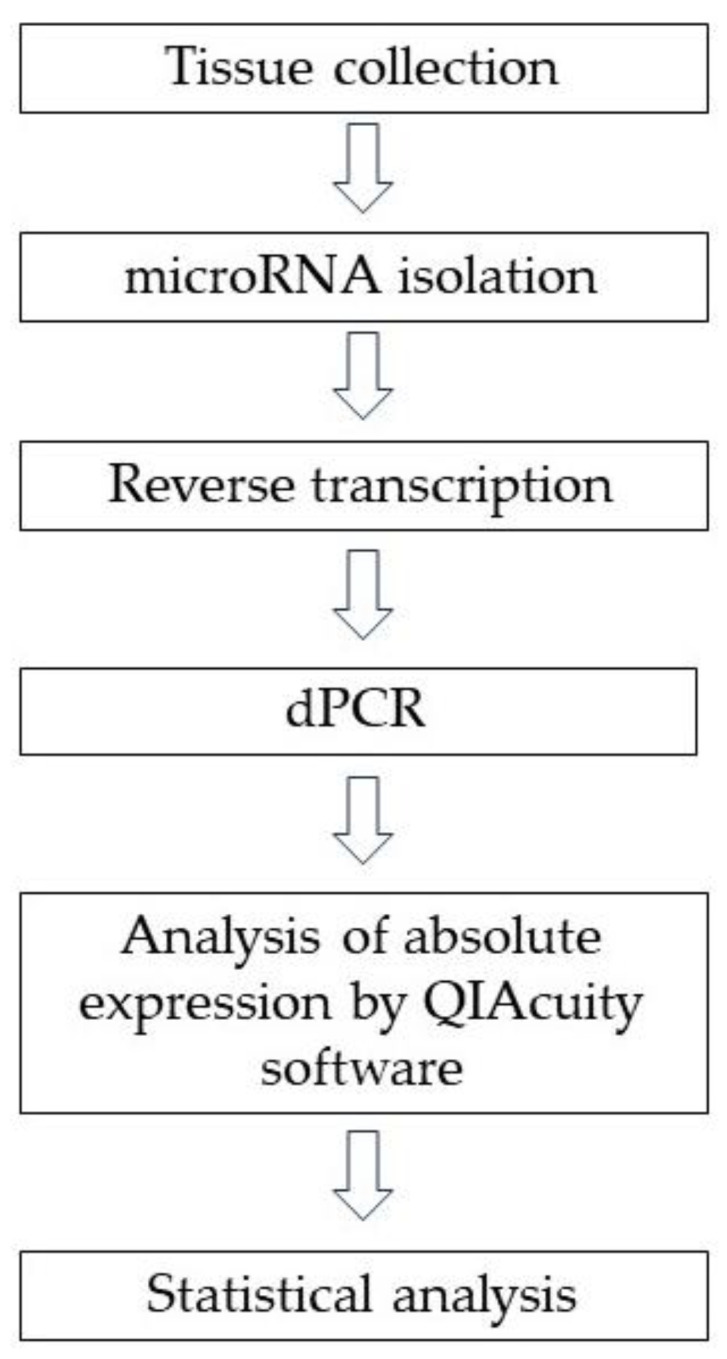
Diagram showing the steps in our research.

**Table 1 ijms-25-03286-t001:** Characteristics and clinical data of the study and control groups.

	Study Group, N = 111 ^1^	Control Group, N = 19 ^1^	*p*-Value ^2^
Age	61 (39–88)	48 (42–92)	<0.001
First period	13 (10–18)	14 (11–17)	0.4
Last menstrual period	51 (37–60)	48 (42–61)	0.043
Birth	2 (0–7)	2 (0–3)	0.2
Cesarean section	0 (0–3)	0 (0–3)	0.5
Miscarriages	0 (0–3)	0 (0–4)	0.2
BMI	29.50 (21.30–50.20)	25.39 (20.08–34.19)	<0.001
Hypertension			0.002
No	48 (43%)	16 (84%)	
Yes	63 (57%)	3 (16%)	
DM			0.042
No	90 (81%)	19 (100%)	
Yes	21 (19%)	0 (0%)	
Hypothyroidism			0.024
No	88 (79%)	19 (100%)	
Yes	23 (21%)	0 (0%)	
Hyperthyroidism			>0.9
No	108 (97%)	19 (100%)	
Yes	3 (2.7%)	0 (0%)	

^1^ Median (Range); ^2^ U Mann–Whitney Test, Chi^2^ test with Fisher’s exact test depending on variable. Statistical significance was demonstrated in the following cases: patient’s age, menopausal status (last menstrual period), hypertension, diabetes (DM), hypothyroidism and BMI.

**Table 2 ijms-25-03286-t002:** Absolute expression of miR-21-5p, miR-205-5p, miR-222-3p, U6, and SNORD48 in the endometrial cancer study group and in the control group.

Absolute Expression (Copies/µL)	Study Group N = 111 ^1^	Control Group N = 19 ^1^	*p*-Value ^2^
miR-21-5p	698,576 (56.96–2,858,160)	938,736 (118,688–3,424,496)	0.2
miR-205-5p	4009.2 (64.08–611,824)	181.52 (32.5–7336.80)	<0.001
miR-222-3p	69,752 (819.2–315 144)	112,168 (35,104–421,912)	0.013
U6	19,390.4 (1034.72–3,321,321)	47,264 (9926.4–119,536)	0.074
SNORD48	79 760 (665.6–581,552)	214,400 (37,168–394,592)	<0.001

^1^ Median (Range); ^2^ U Mann–Whitney Test. Statistically significant results were obtained for the expression of miR-205-5p (*p* < 0.001), miR-222-3p (*p* = 0.013) and SNORD48 (*p* < 0.001). The absolute expression of the tested miRNAs is presented in copies per µL.

**Table 3 ijms-25-03286-t003:** Expression of miRNAs in endometrial cancer tissue (N = 111).

miRNA	Study Group Mean ± SD	Control Group Mean ± SD	Fold Change	log2FC	*p*-Value ^1^
miR-21-5	5.73 ±0.72	5.92 ± 0.49	0.97	−0.05	0.198
miR-205-5p	3.74 ± 0.78	2.68 ± 0.71	1.4	0.48	<0.001
miR-222-3p	4.74 ± 0.45	5.03 ± 0.32	0.94	−0.09	0.022
U6	4.35 ± 0.59	4.54 ± 0.33	0.96	−0.06	0.177
SNORD48	4.85 ± 0.44	5.3 ± 0.25	0.92	−0.12	<0.001

^1^ The comparisons were performed using independent *t*-test and U Mann–Whitney test depending on distribution adjusted *p* value using the Benjamin–Hochberg method. The expression level of miRNA in the tested endometrial cancer tissue is presented in the form of logarithmic data. It was shown that miR-205-5p was upregulated, while miR-222-3p was downregulated. SNORD48 showed statistically significant changes in expression.

**Table 4 ijms-25-03286-t004:** Univariate logistic regression analysis of factors and miRNAs affecting endometrial cancer (N = 111).

	OR (Odds Ratio)	95% CI Lower	95% CI Upper	*p*-Value
Age	1.12	1.06	1.20	<0.001
First period	0.88	0.65	1.19	0.405
Last menstrual period	1.12	0.995	1.26	0.061
Number of Births	1.36	0.92	2.09	0.140
At least one birth	1.37	0.41	3.996	0.585
More than one birth	1.598	0.59	4.29	0.349
More than two births	3.29	0.87	21.54	0.125
Number of Cesarean sections	0.76	0.41	1.59	0.41
At least one CS	0.68	0.21	2.596	0.532
More than one CS	0.57	0.13	4.05	0.508
Number of miscarriages	0.61	0.31	1.24	0.14
At least one miscarriage	0.51	0.17	1.73	0.244
BMI	1.23	1.096	1.42	0.001
Normal BMI (20–25)	0.16	0.05	0.47	<0.001
Obesity (BMI >= 30)	5.05	1.57	22.597	0.0137
Hypertension	7	2.18	31.33	0.003
Smoking	0.24	0.04	1.89	0.128
miR-21-5p	0.48	0.11	1.28	0.263
miR-205-5p	7.27	2.96	22.4	<0.001
miR-222-3p	0.12	0.02	0.53	0.011
U6	0.56	0.24	1.31	0.178
SNORD48	0.04	0.008	0.24	<0.001

Important clinical factors associated with the development of EC are age, BMI, and hypertension. Among the tested miRNAs, two showed a significant association with endometrial cancer: miR-205-5p (OR 7.27, 95% CI: 2.96–22.4, *p* < 0.001) and miR-222-3p (OR 0.12, 95% CI: 0.02–0.53, *p* = 0.011). The study yielded statistically significant results for SNORD48 (OR 0.04, 95% CI: 0.008–0.24, *p* < 0.001).

**Table 5 ijms-25-03286-t005:** Multivariate logistic regression analysis of factors and miRNAs associated endometrial cancer (N = 111).

	OR (Odds Ratio)	95% CI Lower	95% CI Upper	*p*-Value
Age	1.01	0.93	1.1	0.739
Last menstrual period	0.98	0.78	1.22	0.836
BMI	1.16	0.97	1.4	0.11
Hypertension	2.76	0.38	20.05	0.314
miR-205-5 (log)	6.05	1.87	19.59	0.003
miR-222-3 (log)	2.76	0.38	20.05	0.264
SNORD48 (log)	0.01	0.0003	0.47	0.018

It has been shown that miR-205-5p (OR 6.05, 95% CI: 0.38–19.59, *p* = 0.003) is associated with endometrial cancer. This association also held for SNORD48 (OR 0.01, 95% CI: 0.0003–0.47, *p* = 0.018).

**Table 6 ijms-25-03286-t006:** Absolute expression of miR-21-5p, miR-205-5p, miR-222-3p, U6, and SNORD48 in individual degrees of EC differentiation and EIN.

Absolute Expression (Copies/µL)	Control Group N = 19 ^1^	EIN N = 22 ^1^	G1 N = 30 ^1^	G2 N = 47 ^1^	G3 N = 12 ^1^	*p*-Value ^2^
miR-21-5p	938,736 (118,688–3,424,496)	559,560 (177,664–2,858,160)	630,336 (39,392–2,333,184)	711,920 (115.04–2,591,056)	860,536 (56.96–2,264,720)	0.522
miR-205-5p	399.8 (32.5–7337)	1590 (155.5–45,008)	3367.2 (246.72–99,696)	8067 (64.1–113,472)	20,360 (1858.6–94,824)	<0.001
miR-222-3p	112,168 (35,104–421,912)	74,548 (7464–224,816)	83,720 (819.2–261,680)	63,376 (4096.8–315,144)	43,648 (13,256–117,560)	0.058
U6	47,264 (9926.4–119,536)	19,032 (3761.6–113,680)	16,482.4 (1283.84–134,320)	16,656 (1034.72–3,321,312)	133,008 (14,384–590,000)	<0.001
SNORD48	214,400 (37,168–394,592)	97,088 (18,800–304,272)	59,520 (665.6–493,216)	49,520 (11,449.6–459,344)	227,200 (9840–581,552)	<0.001

^1^ Median (Range); ^2^ Kruskal–Wallis rank sum test. Analysis using the Kruskal–Wallis test showed a significant overall difference in the concentration of U6, SNORD48, and miR-205-5, between the groups. Statistical significance was demonstrated for miR-205-5p (*p* < 0.001), miR-222-3p (*p* = 0.058), U6 (*p* < 0.001), and SNORD48 (*p* < 0.001).

**Table 7 ijms-25-03286-t007:** Univariate logistic regression analysis of factors and miRNAs affecting EIN (N = 22).

miRNA	OR	95% CI Lower	95% CI Upper	*p*-Value
miR-21-5	0.48	0.11	1.28	0.263
miR-205-5p	0.48	0.25	0.86	0.018
miR-222-3p	1.74	0.59	6.12	0.35
U6	0.8	0.34	1.8	0.594
SNORD48	1.48	0.51	4.76	0.492

Statistical significance in EIN was achieved for miR-205-5p (OR 0.48).

**Table 8 ijms-25-03286-t008:** Univariate logistic regression analysis of factors and miRNAs affecting G3 histological grade (N = 12).

miRNA	OR	95% CI Lower	95% CI Upper	*p*-Value
miR-21-5p	0.49	0.26	0.91	0.019
miR-205-5p	3.04	1.14	9.73	0.037
miR-222-3p	0.49	0.15	1.84	0.26
U6	9.67	2.96	41.97	<0.001
SNORD48	5.69	1.09	38.01	0.053

Statistical significance in G3 was achieved for miR-21-5p (OR 0.49), miR-205-5p (OR 3.05), U6 (OR 9.67), and SNORD48 (OR 0.48).

**Table 9 ijms-25-03286-t009:** Expression of miR-21-5p, miR-205-5p, miR-222-30, U6, and SNORD48 depending on age only in the study group.

Absolute Expression (Copies/µL)	Age < 50, N = 17 ^1^	Age > 50, N = 94 ^1^	*p*-Value ^2^
miR-21-5p	764,944 (177,664–2,061,856)	698,576 (56.96–2,858,160)	0.509
miR-205-5p	1590 (155.5–113,472)	5057.6 (64.08–611,824)	0.041
miR-222-3p	42,850.7 (7464–223,040)	78,764 (819.2–315,144)	0.063
U6	26,192 (3761.6–107,856)	19,167.2 (1034.72–3,321,312)	>0.9
SNORD48	77,744 (14,462.4–184,144)	80,904 (665.6–581,552)	0.6

^1^ Median (Range); ^2^ U Mann–Whitney Test. We obtained a statistically significant result in the case of miR-205-5p (*p* = 0.041).

**Table 10 ijms-25-03286-t010:** Division of patients depending on body mass index (BMI).

BMI	Study Group N = 111 ^1^	Control Group N = 19 ^1^	*p*-Value ^2^
Normal	14 (13%)	9 (47%)	<0.001
Overweight	43 (39%)	7 (37%)	
Obesity	54 (49%)	3 (16%)	

^1^ n (%); ^2^ Chi^2^ test.

**Table 11 ijms-25-03286-t011:** Division of patients examined in various degrees of differentiation depending on BMI.

BMI	EIN N = 22 ^1^	G1 N = 30 ^1^	G2 N = 47 ^1^	G3 N = 12 ^1^	*p*-Value ^2^
Normal	3 (14%)	3 (10%)	5 (10.6%)	3 (18.2%)	0.72
Overweight	10 (45%)	11 (37%)	16 (34%)	6 (54.5%)	
Obesity	9 (41%)	16 (53%)	26 (55.3%)	3 (27.3%)	

^1^ n (%); ^2^ Chi^2^ test.

**Table 12 ijms-25-03286-t012:** Expression of miR-21-5p, miR-205-5p, miR-222-30, U6, and SNORD48 depending on BMI only in the study group.

Absolute Expression (Copies/µL)	Normal N = 14 ^1^	Overweight N = 43 ^1^	Obesity N = 54 ^1^	*p*-Value ^2^
miR-21-5p	1,092,576 (197,296–2,264,720)	694,496 (56.96–2,858,160)	631,112 (114.08–2,591,056)	0.088
miR-205-5p	6002 (259.5–96,128)	4009.2 (155.52–113,472)	4209.6 (64.08–611,824)	0.869
miR-222-3p	69,112 (4460.8–315,144)	69,752 (7088.8–261,680)	66,988 (819.2–233,960)	0.728
U6	30,168 (5124.8–314,608)	18,704 (1283.84–3,321,312)	18,936 (1034.72–388,256)	0.421
SNORD48	94,232 (26,272–354,656)	91,632 (665.6–581,552)	64,048 (6481.6–519,488)	0.473

^1^ Median (Range); ^2^ Kruskal–Wallis test. No statistically significant results were obtained.

**Table 13 ijms-25-03286-t013:** Expression of miR-21-5p, miR-205-5p, miR-222-3p, U6, and SNORD48 depending on hypertension (HA) only in the study group.

Absolute Expression (Copies/µL)	HA−, N = 48 ^1^	HA+, N = 63 ^1^	*p*-Value ^2^
miR-21-5p	747,056 (115.04–2,858,160)	697 472 (56.96–2,591,056)	0.875
miR-205-5p	3674.8 (64.08–113,472)	4 379.6 (0–611,824)	0.421
miR-222-3p	73,768 (4460.8–223,040)	63,240 (819.2–315,144)	0.554
U6	19,104 (1283.84–590,000)	19,390.4 (1034.72–3,321,312)	0.52
SNORD48	83,224 (665.6–354,656)	77,488 (6481.6–581,552)	0.882

^1^ Median (Range); ^2^ U Mann–Whitney test. No statistically significant results were obtained.

**Table 14 ijms-25-03286-t014:** Expression of miR-21-5p, miR-205-5p, miR-222-3p, U6, and SNORD48 depending on diabetes (DM) only in the study group.

Absolute Expression (Copies/µL)	DM−, N = 90 ^1^	DM+, N = 21 ^1^	*p*-Value ^2^
miR-21-5p	631,728 (56.96–2,858,160)	799,984 (161,648–2,210,368)	0.579
miR-205-5p	3855.2 (64.08–611,824)	8424 (692.24–92,712)	0.164
miR-222-3p	71,460 (4096.8–315,144)	59 248 (819.2–261,680)	0.44
U6	18,816 (1034.72–3,321,312)	24,624 (3926.4–134,320)	0.472
SNORD48	86,920 (665.6–581,552)	58,704 (6481.6–493,216)	0.401

^1^ Median (Range); ^2^ U Mann–Whitney test. No statistically significant results were obtained.

**Table 15 ijms-25-03286-t015:** Expression of miR-21-5p, miR-205-5p, miR-222-3p, U6, and SNORD48 depending on hypothyroidism only in the study group.

Absolute Expression (Copies/µL)	Hypothyroidism−, N = 88 ^1^	Hypothyroidism+, N = 23 ^1^	*p*-Value ^2^
miR-21-5p	680,912 (56.96–2,858,160)	740,480 (166,528–2,210,368)	0.746
miR-205-5p	3894.8 (64.08–108,664)	4549.6 (692.24–611,824)	0.16
miR-222-3p	78,140 (4096.8–315,144)	49,504 (819.2–224,816)	0.192
U6	20,520 (1034.72–3,321,312)	18,704 (1964.8–134,320)	0.859
SNORD48	83,056 (665.6–581,552)	62,112 (6481.6–493,216)	0.178

^1^ Median (Range); ^2^ U Mann–Whitney test. No statistically significant results were obtained.

**Table 16 ijms-25-03286-t016:** Single miRNA stability analysis.

miRNA	GeNorm Stability	NormFinder Stability	BestKeeper Stability	Average Stability
U6	0.29	0.07	0.27	0.21
SNORD48	0.29	0.05	0.17	0.17
U6+SNORD48	0.17	0.04	0.11	0.33

The study assessed miRNA stability analysis: miR U6 and SNORD48 using NormiRazor. U6 and SNORD48 were found to be unstable. Stability analysis was performed on single miRNAs and a combination of two miRNAs (U6 and SNORD48). However, none of the miRNAs tested, used alone or in combination, are sufficiently stable in endometrial cancer tissue to be used as a reference miRNA.

## Data Availability

The raw data supporting the conclusions of this article will be made available by the authors on request.

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
