# Peer review of "Absolute Quantification of Selected microRNAs Expression in Endometrial Cancer by Digital PCR"

_ijms, 2024, doi:10.3390/ijms25063286_

Round 1
Reviewer 1 Report
Comments and Suggestions for Authors
The work is devoted to the study of molecular aspects of the development of endometrial cancer. The essence of the work is to determine the level of various microRNAs in biopsy tumor tissue and complex statistical data processing. Also presented is a large review of data obtained from other studies that clarify the importance of measuring the levels of used RNAs in endometrial cancer samples.
Dear authors, there are several questions regarding your work.
The work is very difficult to understand due to endless abbreviations and gene names, which are either deciphered many times or not deciphered at all. Also, many designations are not brought to a uniform form. There are several typos.
The construction of the text of the article and the presentation of the material also raises questions. The introduction contains a large amount of information that is not used further. In this connection, it is proposed to transform the introduction by presenting important information about the mechanisms of action of the necessary RNAs in the form of a diagram and excluding unnecessary information.
Two classifications of cancer are given, however, in the work the authors rely on the third, without describing it either in the introduction or in the materials and methods. Meanwhile, information about the severity of the disease (G1, G2, etc.), and a possible prognosis in comparison with the RNA level would be extremely useful. It would also be helpful if the cases described in this paper were classified according to the molecular classification given in the introduction.
In the introduction, much attention is paid to the protein PTEN, which is not mentioned in the discussion. Whereas from the introduction it is obvious that the studied RNAs affect its expression level, which leads to a more severe course of the disease. The discussion provides a new set of studies that also determined RNA levels in tumor tissue but does not provide a possible mechanism of operation of the studied RNAs, as well as a possible prognosis for patients with high and low RNA levels.
Questions about statistical data processing.
It is necessary to justify the choice of evaluation criteria. The Wilcoxon (or Mann-Whitney) test has several limitations. It is usually used for relatively small samples (20-30 measurements), and it does not always adequately respond to zero values.
In materials and methods, it is necessary to indicate the range of n values that corresponds to significant differences and present it in tables. And use the same test name in materials and methods as in the tables.
The title of paragraph 1B is unacceptable; the title of the paragraph cannot contain only an adjective.
Table captions must be completely self-contained, that is, contain all the necessary information.
The materials and methods talk about 11 patients in the comparison group, the results talk about 19.
At the end of the text, the authors provide a description of the limitations that their work contains. The comparison group does not correspond to the experimental group in many respects. It may be worth refusing to list the most different indicators (weight, diabetes, hypertension), especially since these are interrelated parameters that influence each other. And place emphasis (including in the discussion) on the data obtained by dividing the experimental group by histological type.
Reviewer 2 Report
Comments and Suggestions for Authors
The manuscript studies the association between the expression of three microRNAs (miRNAs) - miR-21-5p, miR-205-5p, and miR-222-3p - in endometrial cancer tissues, along with the stability of expression of SNORD48 and U6 as reference miRNAs for normalization. Tissue samples were obtained from 111 endometrial cancer patients and 19 patients without neoplastic changes as a control group. Digital PCR was used to measure absolute expression levels.
Results showed that miR-205-5p was upregulated, while miR-222-3p and SNORD48 were downregulated in endometrial cancer tissue compared to the control group. The manuscript is comprehensive, and well-written. However, I have the following concerns:
- More statistical validations are required, like adjusted p-value (FDR) using Benjamin Hochberg, |log_2(Fold-change)| for the expression values.
- KEGG pathway analysis and/or GO Enrichment databases could be used to validate the findings.
Reviewer 3 Report
Comments and Suggestions for Authors
- The abstract provides a concise summary of the study's objectives, methods, and major findings. However, it would be helpful to include a sentence or two discussing the clinical relevance or implications of the results. Consider mentioning any potential diagnostic or therapeutic applications that may arise from this research.
- The introduction adequately introduces the topic of endometrial cancer and the potential involvement of CmiRNAs. However, to provide a more comprehensive background, please consider discussing the current limitations in diagnosing and treating endometrial cancer. Highlight the need for novel biomarkers or therapeutic targets, and how the study of CmiRNAs can contribute to addressing these challenges.
- The methods section is well-written and comprehensive, with clear descriptions of the experimental procedures used. However, there are a few areas that need clarification. Firstly, please provide more information on the selection criteria for patient samples, including the stage and grade of endometrial cancer and any additional characteristics considered during sample selection. This information is essential for understanding the clinical relevance of the findings. Secondly, please specify the number of replicates performed for each experiment to establish statistical significance.
- In the discussion section, while discussing the potential roles of CmiRNAs, elaborate on the biological functions of miR-21-5p, miR-205-5p, and miR-222-3p in endometrial cancer. Discuss their involvement in key cellular processes such as cell proliferation, apoptosis, angiogenesis, invasion, and metastasis. Additionally, explore the potential crosstalk between these CmiRNAs and relevant signaling pathways such as the PI3K/AKT, Wnt/β-catenin, and TGF-β pathways.
- Furthermore, provide insight into the clinical implications of these findings. Discuss how the dysregulation of CmiRNAs could serve as potential prognostic or predictive biomarkers for endometrial cancer. Consider the possibility of developing targeted therapies that specifically modulate the expression or function of these CmiRNAs to improve patient outcomes.
- In the conclusion section, reiterate the main findings of the study and their potential implications for endometrial cancer research and clinical practice. However, do not introduce new information or data in the conclusion. Finally, emphasize the significance of further investigation into the roles of CmiRNAs and suggest specific avenues of research that could build upon this study's findings.
General Comments:
1. In the methods section, please provide the source and reference for SNORD48 and U6, as they are mentioned as reference genes for normalization purposes. Additionally, specify the software or method used for data normalization in the analysis.
2. Consider including a flowchart or diagram depicting the experimental design and data analysis process. This visual representation will help readers better understand the study methodology.
4. Provide a more detailed background on endometrial cancer incidence, prevalence, and current treatment strategies to contextualize the significance of the study.
5. Consider potential confounding factors in endometrial cancer, such as age, BMI, and hormone receptor status, and discuss whether these factors were taken into account during the analyses.
6. Include a section on future directions to provide readers with a roadmap for further research possibilities based on the findings of this study.
Comments on the Quality of English LanguageMinor editing of English language required
Reviewer 4 Report
Comments and Suggestions for Authors
1. Please indicate the specificity and sensitivity of all miRNAs used for in the tested endometrial cancer tissue, respectively.
2. Please show the limit of detection of all miRNAs used for in the tested endometrial cancer tissue, respectively.
3. Please discuss the advantages using digit PCR, compare to real-time PCR.
Comments on the Quality of English LanguageNo
Round 2
Reviewer 1 Report
Comments and Suggestions for Authors
Dear authors, the changes made to the text have made it much clearer and removed all the technical questions that I had. However, questions remain regarding the presentation of the presented material and the scientific novelty of the work. As can be seen from the introduction, many studies have shown changes in the level of the studied RNAs in ovarian cancer, the molecular mechanisms of action of these RNAs are known, but the problem that this work is aimed at solving is not clear. The discussion is largely a retelling of the results of this work and providing information that similar work was carried out by other authors. In this connection, it may be worthwhile in the introduction to focus on highlighting the existing problem that makes this work relevant and necessary. And bring the description of the mechanisms of RNA action into the discussion, comparing it with the data obtained, and drawing a conclusion about the practical significance of determining the level of these RNAs in ovarian cancer.
Also of concern is Table 1. The patients and comparison group differ significantly in age. Accordingly, they will differ in menstrual status, since the older the age, the greater the likelihood of menopause. The same applies to body weight. The greater the mass, the higher the likelihood of diabetes and hypertension. These data do not make much sense and raise questions, since the groups are maximally different in phenotypic parameters and cannot be compared in molecular ones. Is there a need to bring this table?
Reviewer 4 Report
Comments and Suggestions for Authors
The manuscript has been significantly improved.
Comments on the Quality of English LanguageNo.
Round 3
Reviewer 1 Report
Comments and Suggestions for Authors
Dear authors, thank you for the changes made. The added text removes all questions and makes the article much more informative. I hope that your research will continue and it will be possible to recruit an adequate comparison group, which will shed light on the mechanisms of carcinogenesis.
Several typos were noticed in the text: line 709 in group G3 has 11 patients, not 12. And in many places the names of microRNAs are written differently. I hope that during the final proofreading they will be brought to the same form.